# Prevalence of *Mycoplasma bovis* Infection in Calves and Dairy Cows in Western Australia

**DOI:** 10.3390/vetsci9070351

**Published:** 2022-07-11

**Authors:** Jully Gogoi-Tiwari, Harish Kumar Tiwari, Nadeeka K. Wawegama, Chintha Premachandra, Ian Duncan Robertson, Andrew David Fisher, Frank Karanja Waichigio, Pete Irons, Joshua W. Aleri

**Affiliations:** 1School of Veterinary Medicine, College of Science, Health, Engineering and Education, Murdoch University, 90 South Street, Murdoch, Perth, WA 6150, Australia; i.robertson@murdoch.edu.au (I.D.R.); p.irons@murdoch.edu.au (P.I.); 2Asia-Pacific Consortium of Veterinary Epidemiology, Sydney School of Veterinary Science, The University of Sydney, Camden, Sydney, NSW 2570, Australia; harish.tiwari@sydney.edu.au; 3Asia Pacific Centre for Animal Health, Faculty of Veterinary and Agricultural Sciences, University of Melbourne, Parkville, Melbourne, VIC 3010, Australia; nadeekaw@unimelb.edu.au (N.K.W.); cqabbelamaha@student.unimeld.edu.au (C.P.); 4College of Veterinary Medicine, Huazhong Agricultural University, Wuhan 430070, China; 5Hubei International Scientific and Technological Cooperation Base of Veterinary Epidemiology, Huazhong Agricultural University, Wuhan 430070, China; 6Animal Welfare Science Centre, Faculty of Veterinary and Agricultural Sciences, University of Melbourne, Melbourne, VIC 3010, Australia; adfisher@unimelb.edu.au; 7Brunswick Veterinary Services, 27 Ommaney Road, Brunswick, WA 6224, Australia; frankkw@hotmail.com; 8Centre for Animal Production and Health, Future Foods Institute, Murdoch University, 90 South Street, Murdoch, Perth, WA 6150, Australia

**Keywords:** calves, dairy cows, *Mycoplasma bovis*, seroprevalence, Western Australia

## Abstract

**Simple Summary:**

*Mycoplasma bovis* is an emerging pathogen of economic and welfare concern for both adult and young cattle. A study was conducted to determine the prevalence of *M. bovis* in adult cows and calves in the southwest region of Western Australia. Nasal swabs and blood samples were collected from the animals and bulk tank milk samples were assessed for both seroprevalence and active infections of *M. bovis* infections in adult cows and calves. The study recorded a high seroprevalence of *M. bovis* in 699 apparently healthy adult lactating cows and 495 young calves on 29 dairy farms. The herd-level seroprevalence was also detected as being higher in both adult lactating cows and calves. No current active infections were recorded on the farms. The female calves and pure Holstein–Friesian animals were found to be twice as likely to be seropositive for *M. bovis* compared to male calves and the Holstein–Friesian crossbred calves. The high seroprevalence of *M. bovis* in both adult and young cattle in the southwest dairy farms of Western Australia warrants more effective farm biosecurity measures and further evaluation of the current prevention and management measures practiced on the farms.

**Abstract:**

*Mycoplasma bovis* (*M. bovis*) can cause a multitude of diseases in cattle, with detrimental effects on the farm economy and the welfare of both adult and young cattle. The objective of this study was to determine the prevalence of *M. bovis* in adult cows and calves in the south-west region of Western Australia. A cross-sectional study was conducted on 29 dairy farms with 699 apparently healthy adult lactating cows and 495 young calves during 2019–2020. Nasal swabs and blood samples collected from the animals and bulk tank milk (BTM) samples were assessed for *M. bovis*-specific proteins and antibodies by using polymerase chain reaction (PCR) and *Mycoplasma* immunogenic lipase A- Enzyme-Linked Immune Sorbent Assay (MilA ELISA). A seroprevalence of 42.5% (95% CI: 38.9–46.2) and 61% (95% CI: 56.6–65.2) was found in adult lactating cows and calves, respectively. The herd-level seroprevalence of *M. bovis* ranged from 4% (95% CI: 07–19.5) to 92% (95% CI: 75.0–97.8) in adult lactating cows and 25% (95% CI: 10.2–49.5) to 87% (95% CI: 67.9–95.5) for calves in these farms. None of the BTM and nasal swab samples were positive for *M. bovis*, indicating an absence of any current active infections on the farms. The female calves and pure Holstein–Friesian animals are twice as likely to be seropositive for *M. bovis* compared to male calves (OR 2.4; 95% CI: 1.7–3.5) and Holstein–Friesian crossbred calves (OR 2.4; 95% CI: 1.7–3.5). The high seroprevalence in both adult and young cattle in the southwest dairy farms of Western Australia warrants more effective farm biosecurity measures and further evaluation of the current prevention and management measures practiced on the farms.

## 1. Introduction

*Mycoplasma bovis* is a complex, costly, and often overlooked contagious pathogen in dairy cattle [1,2]. It causes mastitis, respiratory disease, conjunctivitis, otitis media, arthritis, and a variety of other conditions and has a detrimental impact on the welfare and productivity of dairy cattle, causing substantial economic losses in the dairy industry [3]. Arthritis, otitis media, and respiratory diseases in adult cattle caused by *M. bovis* are usually associated with mastitis [4,5]. Mastitis in cows caused by *M. bovis* involves multiple quarters resulting a marked decrease in milk production, with a typical lack of response to treatment [6,7]. Once the infection enters a dairy herd, it gets established by rapidly spreading between animals from one quarter to the others through milkers’ hands, contaminated milking machines, or other accessories associated with milking [3,8]. *M. bovis* is the most isolated pathogen from the bulk milk tank (BTM) of herds with clinical or subclinical mastitis [9,10]. Increased herd size has a direct correlation with the detection of mycoplasma infection [11,12], possibly owing to the difficulty in husbandry and management practices, higher animal density, and the movements of animals. Furthermore, mastitis infection rates due to other etiological agents including both Gram-positive and Gram-negative organisms increase in those cows previously infected with *M. bovis* [13,14]. Calves infected with *M. bovis* are more prone to clinical arthritis [4,15], calf pneumonia [16,17], and otitis media [17,18]. They predominantly get infected through contaminated milk from their dams [17], aerosolization of nasal secretions, and nose-to-nose contact with infected animals [3,19].

The major concern associated with *M. bovis* is that it is a highly contagious, rapidly spreading pathogen, and its eradication is difficult once it is established in a herd. The biggest challenge for the control of *M. bovis* infections on the farm is presented by carrier animals. These animals have the potential to disseminate the pathogen to other animals readily without developing the clinical form of the disease [1]. Due to increasing resistance against a variety of antimicrobial agents and the absence of an effective vaccine, this pathogen is of worldwide concern [20]. Early detection of the pathogen in sera, milk, and nasal swab samples is critical to plan and adopt appropriate control measures on farms in the absence of an effective vaccine [20,21]. Additionally, adopting appropriate farm biosecurity measures to prevent the entry of the pathogen is critical for non-infected herds. 

*M. bovis* infection in Australia was first detected in 1970 from bovine milk samples [22] and since then the presence of the pathogen has been confirmed in all the states and territories of Australia [23]. The herd-level prevalence of *M. bovis* in dairy cattle as isolated from BTM in Victoria and North Queensland in Australia was reported to be 43–62% in herds with somatic cell count (SCC) > 250 × 10^3^ cells/mL [3,24]. However, a follow-up study in 2014 reported a herd-level prevalence of *Mycoplasma* infection as low as 0.9% [23] in Australia. Although both the studies used PCR assays for detection of the herd-level prevalence of *M. bovis*, the earlier study did not assess the diagnostic sensitivity and specificity of the test, which might have affected the analytical sensitivity and specificity of the test resulting in a big difference. Despite *M. bovis* being prevalent in Australia for the last five decades, limited research has been undertaken to explore the prevalence of the pathogen in dairy herds [23,24]. In 2010–2012, a quantitative reverse transcription polymerase chain reaction assay (qPCR) analysis of nasal swabs collected from apparently healthy cattle before their live export reported a 4.8% prevalence of *M. bovis* in Western Australia [25]. However, to the best of our knowledge, no studies have been conducted in Western Australian dairy herds to explore the prevalence of *M. bovis*. The current study was undertaken to investigate the prevalence of *M. bovis* in (i) dairy cows and (ii) calves in the dairy farms in Western Australia by using both serological and PCR methods. A combination of PCR and serological methods will not only identify the presence of *M. bovis* on the farms but will also reveal if the animals were previously infected with *Mycoplasma*. The data generated on the prevalence of *M. bovis* from this study will contribute toward on-farm biosecurity risk assessment and help the dairy farmers adopt efficient on-farm management practices to prevent and control the pathogen and to reduce its impact on the morbidity and mortality of their animals.

## 2. Material and Methods

### 2.1. Ethics Approval 

The study was conducted in accordance with the Australian Code of Practice for the Care and Use of Animals for Scientific Purposes 2013, with the approval of the Human Research and Animal Ethics Committees of Murdoch University, Approval No. R3144/19 and 2019/047, respectively. 

### 2.2. Study Area

The study was conducted in the south-west region of Western Australia (Figure 1). The region has a temperate Mediterranean climate with an annual rainfall of approximately 730 mm. Dairy farms are predominantly located southwest of Perth (capital city of Western Australia), being well-suited to pasture-based feeding systems.

### 2.3. Study Design and General Data Collection

Data were collected from April 2019–June 2020. This was a cross-sectional study where study farms were visited once to obtain blood samples from healthy calves ≤7 days old and from adult lactating cows. 

### 2.4. Study Populations 

A total of 29 farms participated in the study. A convenience sample of 140 registered dairy producers were invited via email. Additional expressions of interest to the dairy producers were sent via a regional newsletter (Feed Trough) and during a regional farmer’s day event. Participation was voluntary, and no incentives were provided. A two-stage cluster sampling technique was utilised to select study subjects. In farms with less than 25 calves at the time of sampling, all calves were included in the study. In those farms that had more than 25 calves in the required age group, a total of 25 calves were included in the study. The target population included healthy calves ≤7 days old and adult lactating dairy cows reared in a Mediterranean pasture-based production in Australia. 

### 2.5. Sample Collection

#### 2.5.1. Blood 

A total of 1194 blood samples (699 from cows and 495 from calves) were collected. Blood samples were collected by jugular and tail venipuncture for calves and adult animals, respectively into sterile serum separator vacutainer serum tubes (Becton Dickinson and Company, Belliver Industrial Estate, Belliver Way, Roborough, Plymouth PL6 7BP UK). Blood samples were allowed to clot at ambient temperature and then transported on ice to the Production Animal Medicine Laboratory at Murdoch University and serum was separated by centrifugation for 15 min at 700× *g* (at room temperature). Sera samples were stored at −80 °C until tested. 

#### 2.5.2. Milk

A bulk tank milk (BTM) sample was collected from each of the 29 dairy farms involved in the study. A total of 100 mL of milk samples were collected aseptically from the bulk milk tank and then transported on ice to the Production Animal Medicine Laboratory at Murdoch University. The samples were stored at −80 °C until further laboratory analysis.

#### 2.5.3. Nasal Swabs

A total of 495 nasal swab samples were collected from the healthy calves from the 29 dairy herds. The calves were restrained in a head bail and sampled by using modified McCullough–Cartwright large animal double-guarded culture swabs made of two rigid concentric tubes, with the culture swab in the center (Har-Vet, West Bend, WI, USA). These were inserted approximately 10–15 cm into the nasal cavity and the swab rotated across the nasal mucosa to collect a sample of the nasal secretions. The swabs were then immersed in 0.5 mL of RNAlater^®^ (Sigma-Aldrich, St. Louis, MO, USA) and stored at −20 °C until further investigation. 

### 2.6. Laboratory Analysis

#### 2.6.1. Enzyme-Linked Immunosorbent Assay (ELISA) 

A total of 1194 sera and 29 bulk tank milk (BTM) samples were analyzed with an in-house ELISA-MilA ELISA [26]. This ELISA detects the presence of antibodies to *M. bovis* in serum and milk, and the procedure followed has been described previously [2,26]. Briefly, a 96-well Nunc Maxisorp plate (Thermofisher Scientific, Scoresby, VIC, Australia) was coated with glutathione *S*-transferase (GST)-MilA-Ag by using 1.2 µg per well. Test serum, positive and negative controls [26] were diluted by using phosphate-buffered saline containing 0.05% [vol/vol] Tween 20 (PBS-T). Serum samples were diluted to 1/300. Milk samples were diluted at 1/50. Each sample was tested in duplicate. HRP-conjugated anti-bovine sheep antibody (Bethyl Laboratories, Inc, Montgomery, TX, USA) and 2,2′-azino-bis (3-ethylbenzothiazoline-6-sulfonic acid) (ABTS) were used as conjugate and enzyme-substrate, respectively. Plates were incubated for 7 min at room temperature and the reaction was stopped by using 1% SDS. The absorbance was read at an optical density of 405 nm by using a microplate reader (Bio-Rad Laboratories, South Granville, NSW, Australia). The cut-off value used for interpreting the samples was 140 antibody units (AU), as suggested by Wawegama and co-workers [2]. All the optical density data generated in the MilA ELISA were analysed by using an online ELISA analysis programme (http://www.elisaanalysis.com accessed on 21 December 2021).

#### 2.6.2. DNA Extraction and *M. bovis* Quantitative PCR

The DNA from nasal swabs was extracted by using a MagMAX CORE Nucleic Acid Purification Kit (Thermo Fisher Scientific, Waltham, MA, USA) according to the manufacturer’s instructions. The DNA was used as the template for an *M. bovis*-specific quantitative PCR [27] by using the Rotor-Gene Q thermocycler (Qiagen, Clayton, VIC, Australia), and the *M. bovis* oppD gene ligated into commercially available plasmid pGEM-T according to the manufacturer’s recommendations (Promega, Madison, WI, USA) was used for the standard curve. 

### 2.7. Statistical Analysis 

All statistical analyses were performed in the R programming environment [28]. The R packages “oddsratio”, “prevalence”, and “ggplot2” for calculation of chi-square tests, prevalence, and for plots and graphs were used to analyse the data, respectively.

## 3. Results

### 3.1. Seroprevalence of M. bovis in Apparently Healthy Adult Lactating Cows

The overall seroprevalence of *M. bovis* in 699 healthy adult lactating cows from 29 farms in the southwest region of Australia was 42.5% (95% CI: 38.9–46.2) (Table 1). The farm-level seroprevalence ranged from 4% (95% CI:07–19.5) for farms 12 and 28 to 92% (95% CI: 75.0–97.8) for Farms 4 and 8 (Table 1).

### 3.2. Seroprevalence of M. bovis in Apparently Healthy Calves

The overall seroprevalence in 495 apparently healthy calves from the 29 farms was 61.0% (95% CI: 56.6–65.2) (Table 2). The farm-level seroprevalence ranged from 25% (95% CI: 10.2–49.5) to 87% (95% CI: 67.9–95.5) for Farms 20 and 2, respectively (Table 2).

### 3.3. ELISA-Based Detection for M. bovis-Specific Antibodies in Bulk Milk Samples

All the 29 BTM samples assessed for *M. bovis*-specific antibodies were found to be negative on the ELISA.

### 3.4. PCR Detection for M. Bovis-Specific Proteins in Nasal Swab Samples of Apparently Healthy Calves

All the 295 nasal swab samples collected from apparently healthy calves were found to be negative for *M. bovis*-specific proteins in PCR test.

### 3.5. Intrinsic Factors Associated with M. bovis Seroprevalence in Cows and Calves 

The odds of a female calf being seropositive was 2.4 (95% CI: 1.7–3.5) that of males (*p* < 0.001) (Table 3). Similarly, the odds of Holstein Friesian calves being seropositive was 2.4 (95% CI: 1.7–3.5) that of Holstein–Friesian cross calves (*p* < 0.001) (Table 3).

## 4. Discussion

*M. bovis* is an emerging pathogen of economic and welfare concern for both adult and young cattle [29]. It usually gets introduced to the farm through the introduction of new animals that are sub-clinical carriers. Once the pathogen enters a farm, it is nearly impossible to prevent it from spreading between animals due to its highly contagious nature [30]. Furthermore, the pathogen possesses multiple virulence factors [31] and innate resistance ability [27,31] to different antimicrobial agents, making treatment challenging [32]. In the absence of effective treatments or vaccines that give unequivocal results, effective farm biosecurity, prevention, and management measures are pivotal. This cross-sectional study was undertaken to determine the prevalence of *M. bovis* infections in dairy animals in Western Australia by using both ELISA and PCR. ELISA and PCR are the most used diagnostic tests for the detection of *M. bovis* in serum, milk, and other associated clinical samples [21,33,34]. However, unlike PCR, an antibody ELISA possesses the additional advantages of being able to detect infections in those animals that were exposed to the pathogen weeks earlier [35,36]. In the current study, high animal and herd seroprevalences to *M. bovis* among both adult lactating cows and calves were recorded (42.5% and 61.0%, respectively). 

The farm-level seroprevalence of *M. bovis* in adult lactating cows ranged from 4 to 92% in the 29 dairy farms sampled in the southwest region of Western Australia. None of these lactating animals were found to be positive for active infections as all the 29 BTM samples were negative on the ELISA. Similarly, the farm-level seroprevalence of *M. bovis* in calves on those farms was also high ranging from 25 to 87%. Active infections were also not detected in the nasal swabs sampled from these animals (negative PCR test). The high prevalence of *M. bovis* antibodies in the serum samples of lactating cows was most likely due to previous exposure to *M. bovis*, whereas in calves it may be due to the transfer of passive immunity via colostrum. Furthermore, the better sensitivity of MilA ELISA test compared to other available ELISA tests resulted in the detection of all those seropositive animals against *M. bovis* despite the fact that the organism might have possibly been cleared from their bodies [26,37]. The negative BTM and nasal swab samples indicate the lack of current *M. bovis* infections on the farms. 

In the current study, female calves were found to be twice as likely (OR 2.4; 95% CI:1.7–3.5) to become seropositive with *M. bovis* as the male calves. Previous studies conducted in the United States and Canada [38,39] reported higher case fatality rates in female calves. Loneragan and co-workers’ study on 21-million feedlot cattle in the United States reported higher mortalities in heifer cattle due to bovine respiratory disease than the steers. In contrast, few studies [40,41,42,43] including an Australian one [44] reported the higher risk of bovine respiratory disease in male calves compared to the female calves. Similarly, Holstein–Friesian calves were found to be more likely (OR 2.4; 95% CI: 1.7–3.5) to be *M. bovis*-seropositive than Holstein–Friesian cross-bred calves. Breed difference was cited as an important factor for susceptibility to respiratory infections in animals [40,45]. Fewer *M. bovis* infections in Holstein–Friesian cross may be attributed to the beneficial characteristics of hybrid vigour in such cross-bred cattle [45]. 

Serological methods, including ELISAs, have been shown to be reliable and effective diagnostic tools in assessing the biosecurity risk of *M. bovis* infection in animals prior to their introduction to a herd. It is prudent to also consider the limitations when using these tools for diagnostic purposes. First, although a MilA ELISA detects the antibodies present against *M. bovis* in animals, this does not imply that the animals are currently infected or infectious. Secondly, further knowledge on the seroconversion duration and longevity of the *M. bovis*-specific antibodies are crucial in the current context of serological diagnosis of *M. bovis* infections by using ELISA. Future research is required to explore the animal level and herd-level risk factors of *M. bovis* infections to fully understand the potential routes of entry of the pathogen to the dairy farms and to understand the knowledge, attitudes, and practices (KAP) of the dairy farmers about *M. bovis* in Western Australia. The potential information generated from this study may be of interest to the dairy farmers and the relevant departments as this information may aid the stakeholders in adopting appropriate future strategies for the prevention and management of *M. bovis* infections. 

## 5. Conclusions

This study reports seroprevalence of *M. bovis* exposures in the dairy farms of the southwest region of Western Australia. The high seroprevalence of *M. bovis* in both adult and young animals in Western Australian dairy farms is potentially a matter of concern. Attention should be given to farm biosecurity measures to prevent and manage the entry and spread of the pathogen on the farm. Screening of all new animals should be undertaken prior to their introduction to a negative farm.

## Figures and Tables

**Figure 1 vetsci-09-00351-f001:**
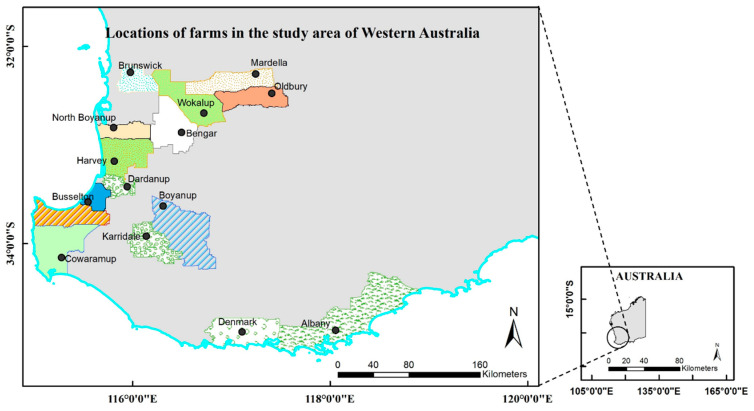
Locations of the farms in Western Australia for the *Mycoplasma bovis* seroprevalence study.

**Table 1 vetsci-09-00351-t001:** Seroprevalence of *Mycoplasma bovis* exposures in adult animals in different farms in the southwest region of Western Australia.

Farm ID	Adults	HF *	HFX **	Positive	Prevalence (95% CI)
F1	25	25	0	18	72.0 (52.4–85.7)
F2	25	25	0	19	76.0 (57.0–89.0)
F3	18	18	0	5	27.8 (12.5–50.9)
F4	25	25	0	23	92.0 (75.0–97.8)
F5	24	24	0	22	91.7 (74.1–97.7)
F6	25	0	25	21	84.0 (65.3–93.6)
F7	25	25	0	22	88.0 (70.0–95.8)
F8	25	25	0	23	92.0 (75.0–97.8)
F9	25	0	25	19	76.0 (56.6–88.5)
F10	22	0	22	5	22.7 (10.1–43.4)
F11	25	0	25	3	12.0 (4.2–30.0)
F12	25	0	25	1	4.0 (0.7–19.5)
F13	24	0	24	3	12.0 (4.2–30.0)
F14	25	0	25	4	16.0 (6.4–34.6)
F15	25	1	24	3	12.0 (4.2–30.0)
F16	25	0	25	10	40.0 (23.4–59.3)
F17	25	24	1	3	12.0 (4.2–30.0)
F18	24	24	0	3	12.5 (4.3–31.0)
F19	25	25	1	5	20.0 (8.8–39.1)
F20	25	25	0	7	28.0 (14.3–47.6)
F21	25	25	0	19	76.0 (56.6–88.5)
F22	23	23	0	3	13.0 (4.5–32.1)
F23	21	21	0	4	19.0 (7.7–40.0)
F24	25	25	0	3	12.0 (4.2–30.0)
F25	23	23	0	9	39.1 (22.2–59.2)
F26	22	22	0	15	68.2 (47.3–83.6)
F27	24	24	0	17	70.8 (50.8–85.1)
F28	25	25	0	1	4.0 (0.7–19.5)
F29	24	24	0	7	29.2 (14.9–49.2)
**Total**	**699**	**475**	**222**	**297**	**42.5 (38.9–46.2)**

* HF Holstein–Friesian; ** HFX Holstein–Friesian cross.

**Table 2 vetsci-09-00351-t002:** Prevalence of *Mycoplasma bovis* exposures in calves in different farms in the southwest region of Western Australia.

Farm ID	Calves	HF *	HFX **	Positive	Prevalence (95% CI)
F1	23	20	3	15	65.2 (44.9–81.2)
F2	25	19	6	17	68.0 (48.4–82.8)
F3	25	25	0	21	84.0 (65.3–93.6)
F4	6	2	4	3	50.0 (18.7–81.2)
F5	25	19	6	11	44.0 (26.7–62.9)
F6	25	0	25	9	36.0 (20.2–55.5)
F7	25	25	0	19	76.0 (56.6–88.5)
F8	5	0	5	3	60.0 (23.1–88.2)
F9	23	0	23	15	65.2 (44.9–81.2)
F10	22	0	22	9	40.9 (23.3–61.3)
F11	25	14	11	20	80.0 (60.9–91.1)
F12	18	18	0	10	55.6 (33.7–75.4)
F13	16	0	16	8	50.0 (28.0–72.0)
F14	14	2	12	10	71.4 (45.3–88.3)
F15	25	0	25	18	72.0 (52.4–85.7)
F16	21	0	21	16	76.2 (54.9–89.4)
F17	25	23	2	12	48.0 (30.0–66.5)
F18	20	20	0	7	35.0 (18.1–56.7)
F19	14	2	12	9	64.3 (38.8–83.7)
F20	16	15	1	4	25.0 (10.2–49.5)
F21	23	23	0	20	87.0 (67.9–95.5)
F22	14	5	9	12	85.7 (60.1–96.0)
F23	0	-	-	-	-
F24	15	12	3	8	53.3 (30.1–75.2)
F25	13	13	0	9	69.2 (42.4–87.3)
F26	12	0	12	7	58.3 (31.9–80.7)
F27	0	-	-	-	-
F28	0	-	-	-	-
F29	20	15	5	10	50.0 (29.9–70.1)
**Total**	**495**	**272**	**223**	**302**	**61.01 (56.6–65.2)**

* HF Holstein Friesian; ** HFX Holstein Friesian cross.

**Table 3 vetsci-09-00351-t003:** Test of association of intrinsic factors on *M. bovis* positivity by ELISA testing in calves and adults.

		*M. bovis* ELISA		OR (95% CI)	*p* Value
Calves	Total	Negative	Positive		
*Sex*					
Female	278	102	169	2.4 (1.7–3.5)	<0.001
Male	214	90	131	1.0	
**Total**	**492**	**192**	**300**		
*Breed*					
HF	272	106	165	2.4 (1.7–3.5)	<0.001
HFX	223	87	137	1.0	
**Total**	**495**	**193**	**302**		
**Adults**					
*Breed*					
HF	576	331	245	1.0 (0.7–1.5)	0.5
HFX	123	71	52	1.0	
**Total**	**699**	**402**	**297**		

## Data Availability

The original contributions presented in the study are included in the article and further inquiries can be directed to the corresponding authors.

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
