# Peer review of "Prevalence of Mycoplasma bovis Infection in Calves and Dairy Cows in Western Australia"

_vetsci, 2022, doi:10.3390/vetsci9070351_

Round 1
Reviewer 1 Report
I read the manuscript entitled "Prevalence of Mycoplasma bovis infection in calves and dairy cows in Western Australia" with pleasure. However, I would like to make a few remarks.
My main question concerns whether there is any regularity in the spaciotemporal distribution of seropositive animals? That is, does the detected percentage of seropositive cows depend on the region where the farm is located? Also, is the change in the number of seropositive animals related to the sampling season, that is, can this indicator fluctuate from April to June due to seasonal changes in the opportunistic microflora? The same question concerns the reasoning about the difference in the number of seropositive animals between different sex and age groups - does it vary from region to region in the areas studied by the authors? Can this distribution be dependent on other factors, such as the initial sex and age composition of the animals kept on the farm?
Below I list a few minor remarks:
1) Why is "Western Australia" in the title of manuscript written in small print and moved to another line?
2) Introduction section
Should this section be written in italics?
3) Line 81
Maybe it would be better to decipher the abbreviation SCC?
Author Response
Thank you very much for your comments and suggestions. We have revised the manuscript now as per the suggestions. Please refer to the points below for the revision.
1. My main question concerns whether there is any regularity in the spatiotemporal distribution of seropositive animals? That is, does the detected percentage of seropositive cows depend on the region where the farm is located?
We have included a total of 29 farms in the study from the South-West region of Western Australia. The seroprevalence of M.bovis in both adult cows and calves varied on different farms. In the case of adult cows, the seroprevalence of M.bovis ranged from 4-92%, and in calves 25-87% on different farms. There are many factors that could be responsible for the variation of seroprevalence in these farms and undoubtedly location could be one of the factors.
2. Also, is the change in the number of seropositive animals related to the sampling season, that is, can this indicator fluctuate from April to June due to seasonal changes in the opportunistic microflora?
You are absolutely right. The season may play a role in the occurrence of M.bovis on the farm. In our future studies, we would like to explore the effect of season on the seroprevalence of M.bovis on these farms.
3. The same question concerns the reasoning about the difference in the number of seropositive animals between different sex and age groups - does it vary from region to region in the areas studied by the authors? Can this distribution be dependent on other factors, such as the initial sex and age composition of the animals kept on the farm?
In the current study, we found that the female calves were twice likely of becoming seropositive as the male calves ((OR 2.4; 95% CI:1.7-3.5). This was found across the 29 farms investigated. We only included healthy calves ≤ 7 days old in the current study. It will be interesting to study the role of age of these animals in the seroprevalence of M.bovis in future studies.
4. Why is "Western Australia" in the title of manuscript written in small print and moved to another line?
This is a formatting error. We have now revised the manuscript. Please check line numbers 1-2 in the attached revised manuscript.
5. Introduction section- Should this section be written in italics?
This was again a formatting error. We have revised the manuscript now. Please check line numbers 43-106 in the attached revised manuscript.
6. Line 81- Maybe it would be better to decipher the abbreviation SCC?
Thank you for the suggestion. We have now provided the full form of the abbreviation in the text. Please check line number 85 in the attached revised manuscript.

Reviewer 2 Report
The title needs to be reformatted to fit on two lines.
The entire introduction is in italics. Please change typeset to non-italics.
On line 198, "r" should be capitalized
Author Response
Thank you for your comments. We have revised the manuscript now as per your suggestions. Please refer to the changes below.
- The title needs to be reformatted to fit on two lines.
This was a formatting error. This has been addressed now. Please check line numbers 1-2 in the attached revised manuscript.
2. The entire introduction is in italics. Please change the typeset to non-italics.
This was again a formatting error. This has been addressed now. Please check line numbers 43-106 in the attached revised manuscript.
3. On line 198, "r" should be capitalized.
This has been addressed now. Please check line number 204 in the attached revised manuscript.
